# Preparation, Characterization, and Bio Evaluation of Fatty N- Hexadecanyl Chitosan Derivatives for Biomedical Applications

**DOI:** 10.3390/polym14194011

**Published:** 2022-09-25

**Authors:** Hanaa Mansour, Samia El-Sigeny, Sarah Shoman, Marwa M. Abu-Serie, Tamer M. Tamer

**Affiliations:** 1Department of Chemistry, Faculty of Science, Kafrelsheikh University, Kafrelsheikh 33516, Egypt; 2Medical Biotechnology Department, Genetic Engineering and Biotechnology Research Institute (GEBRI), City of Scientific Research and Technological Applications (SRTA-City), Alexandria 21934, Egypt; 3Polymer Materials Research Department, Advanced Technology and New Materials Research Institute (ATNMRI), City of Scientific Research and Technological Applications (SRTA-City), Alexandria 21934, Egypt; 4Infochemistry Scientific Center, ITMO University, 191002 Saint Petersburg, Russia

**Keywords:** N-alkylation, antibacterial activity, chitosan, hexadecane, characterization

## Abstract

The objective of this study was to improve the antibacterial activities of chitosan via N-alkyl substitution using 1-bromohexadecane. Mono and di substitution (Mono-NHD-Ch and Di-NHD-Ch) were prepared and characterized using FT-IR, HNMR, TGA, DSC, and SEM. Elemental analysis shows an increase in the C/N ratio from 5.45 for chitosan to 8.63 for Mono-NHD-Ch and 10.46 for Di-NHD-Ch. The antibacterial properties were evaluated against *Escherichia coli*, *Pseudomonas aeruginosa*, *Staphylococcus aureus*, and *Bacillus cereus*. In the examined microorganisms, the antibacterial properties of the novel alkyl derivatives increased substantially higher than chitosan. The minimum inhibitory concentration (MIC) of Mono-NHD-Ch and Di-NHD-Ch was perceived at 50 μg/mL against tested microorganisms, except for B. cereus. The MTT test was used to determine the cytotoxicity of the produced materials, which proved their safety to fibroblast cells. The findings suggest that the new N-Alkyl chitosan derivatives might be used as antibacterial alternatives to pure chitosan in wound infection treatments.

## 1. Introduction

Chitosan (a deacetylated version of chitin) is one of nature’s polymers commonly derivatized from chitin, and it is second only to cellulose in terms of natural abundance [1]. Chitin and chitosan are both found in crustaceans and insects’ shells and various other species such as fungi, algae, and yeast. Chitosan is a biopolymer made up of glucosamine units and N-acetylated glucosamine. In addition, β-(1-4) glycosidic linkages connect them. Chitosan has been shown to have various biological properties, including antibacterial [2,3], anticancer [4], antioxidant [5], hemostatic action [6], and drug delivery [7], as well as the ability to speed wound healing [8,9].

It is widely known that chitosan has reasonable antibacterial characteristics against various bacteria, and these properties are linked to the amount of chitosan deposition on bacterium cells [10,11]. However, due to the constant changes of bacteria and lower rates of developing new antimicrobial agents, the activity of primary chitosan against microbial infections has become progressively inhibited. Therefore, the chemical transformation of chitosan function groups offers an excellent choice to generate an effective antibacterial against a new generation of bacteria. Furthermore, chemical modification of a biopolymer such as chitosan usually results in new materials with capabilities that allow for a broader range of pharmacy [12,13], food and medicine [14], water treatment [15,16], and remediation applications [17].

Several studies have clearly explained the role of amine groups in antibacterial mechanisms against both Gram-negative and Gram-positive bacteria. Formation of protonated cation derivative simplified the polymers role in distorting the cell wall or penetrating it inside the microorganism cells [18,19]. Functionalization of chitosan with hydrophobic side chain showed interesting improvement in the antibacterial activity [20,21]. Reductive alkylation of chitosan was widely utilized as the easiest way to introduce alkyl groups to the amino group of chitosan [22]. Because reductive alkylation is N-selective and results in mono- or dialkyl-derivatives, avoiding quaternization, it is frequently favored over replacement reactions with alkyl halides.

This research focused on the synthesis and characterization of novel chitosan N-Alkyl derivatives. Furthermore, the antibacterial properties of the materials produced were assessed for usage in biomedical applications.

## 2. Results

Antibiotic resistance in bacteria is prompting scientists to explore new materials to combat the mutations. Therefore, amination, methylation, and other chitosan derivatives were synthesized to improve their antibacterial action [23,24,25]. The following study prepared new N alkyl derivatives of chitosan by interacting with amine groups with 1-Bromohexadecane (as schematically illustrated in Figure 1). Chemical structural changes were monitored by FT-IR, TGA, DSC, and spectroscopic analyses to confirm the production of its N-alkyl chitosan derivative.

### 2.1. Characterization

It is generally recognized that the hydrophilicity of biomaterials has a significant impact on their interactions with living cells. The water absorption values for chitosan and its two alky derivatives, Mono-NHD-Ch and Di-NHD-Ch, are 198.31 ± 9.92, 137.15 ± 6.86, and 104.19 ± 5.21 (%), respectively. The water uptake of the chitosan derivative decreases significantly after coupling with 1-Bromohexadecane as a monosubstituted, whereas it is continuously reduced after di substitution by almost 50%. When amine groups were coupled with 1-Bromohexadecane, hydrophilic sites were substituted with aliphatic long-chain ones, lowering polymer hydrophilicity. The moisture content of chitosan and its alkyl derivatives was tested and calculated as 9.52 ± 0.48 for chitosan, 7.61 ± 0.38 for Mono-NHD-Ch, and 6.18 ± 0.31 for Di-NHD-Ch, indicating their rise in hydrophobic character.

Ion exchange capacity was used to estimate the influence of alkyl substitution on chitosan’s surface free amine groups. Pure chitosan records 5.49 ± 0.27 meq/g compared to 5.25 ± 0.26 meq/g for Mono-NHD-Ch and 4.44 ± 0.22 meq/g for Di-NHD-Ch. The measurements focused on the detection of free amine groups on the surface of membranes. The consumption of free amine groups in new derivatization has a direct effect on polymer ion exchange capacity [26].

The derivative of chitosan with alkyl substitution was also identified by elemental analysis, as presented in Table 1. In the case of chitosan, the significant decrease in the N% ratio in the mono and di derivatization, which was 2.37% and 2.09%, respectively, from 7.17%, indicated the substitution process of the Alky groups in the polysaccharide backbone [27].

Figure 2a shows the usual bands of chitosan function groups [28,29]. The broad band around 3461 cm^−1^ corresponds to a combination of the stretching vibration of the NH2 and OH groups. A band at 2925 cm^−1^ refer to the stretching vibration of the C-H bond; the stretching vibration of methylene groups was present at 2850 cm^−1^. Bands at 1655 cm^−1^ point out the stretching vibration of the C = O and NH-C = O functional groups. The band at 1459 cm^−1^ corresponds to -NH_2_ bending vibration. The peak at 1313 cm^−1^ is ascribed to C-N stretching vibration. Bands at 1076 cm^−1^ correspond to C-O-H group stretching.

The UV-visible spectra of chitosan and its N-alkyl derivatives are investigated in Figure 2b. The apparent absorbance band at a maximum of 290 nm could be attributed to the n-σ * transition of amine-free electrons in the case of chitosan [30]. In the case of mono and di N-alkyl substitutions, however, the rise in peak strength and observed shift to the higher wavelengths of 296 and 310 nm, respectively, could be explained by the alkyl substituent’s increased donor ability to stabilize the excited state [31].

The TGA of chitosan and its alkyl derivatives was examined, and the results are presented in Figure 2c-1. Like most polysaccharides, chitosan demonstrates a three-stage degradation profile. The first stage refers to the loss of moisture associated with its hydrophilic groups (13.01%) [32], while the second stage, with around 38.98% (between 240 and 325 °C), was explained by thermal decomposition of the glucopyranose ring-producing ketonic complex [33], and the final stage at a higher temperature is associated with the decomposition of intermediated ketonic complex [34]. The thermal profile of an alkyl derivative is more complicated as a result of the presence of a substituted alkyl group along the polymer backbone. It was recognized to decrease the moisture content to 7.84 and 6.01% for Mono-NHD-Ch and Di-NHD-Ch, respectively. The second stage (that regraded to the decomposition of the pyranose ring) was limited to being 28.62 for Mono-NHD-Ch and 31.46% for Di-NHD-Ch at almost the same temperature zone as chitosan. The third decomposition stage was divided into alkyl derivatives and a slower part from 325 °C to 540 °C with a weight percent of 17.3% for Mono-NHD-Ch and 20.19% for Di-NHD-Ch, and the faster part at higher temperatures, with weight loss of around 29.7% for both derivatives.

DSC experiments were used to estimate the thermal characterization, and the results are shown in Figure 2c-2. The findings are in good agreement with what has been found in the literature [32,33]. Chitosan revealed an endothermic peak of approximately 100 °C due to the release of moisture content in the polymer that was not completely removed during drying. In addition to the exothermic peak, glucopyranose units break down at 308 °C for chitosan and 263 °C and 258 °C for mono and di substitution. The movement of the peak to a lower temperature showed a change in the chitosan chain structure.

In Figure 2d, SEM micrographs of Ch and its alkyl derivatives are shown. The roughness of the investigated surfaces increased with the Alkyl functionalization of chitosan, as indicated in the micrographs. This is due to the presence of heterogeneous molecules between the polymeric chitosan chains and the immobilization of the alkyl group into glucose amine groups on the repeating polysaccharide, which alters the internal chain order and changes the polymer crystal structure.

Figure 3 demonstrates the HNMR of Ch, Mono-NHD-Ch, and Di-NHD-Ch. The Ch chart demonstrates the amino glucopyranose ring. Signals at δ 4.9 ppm are attributed to the proton in C1, whereas signals at δ 3.00 ppm refer to hydrogen attached to C2. Multiple signals between δ 3.39 and δ 3.73 ppm refer to protons in C3, 4, 5, and 6. The protons of the acetyl groups in chitin monomer residue appeared at δ 1.85 ppm [35]. Substitution of a mono-alkyl side chain on the amine group resulted in a multipeak related to alkyl protons numbered H11 at 2.75 ppm, H12 at 1.3 ppm, and H13 at 0.9 ppm, with the proton of mine hydrogen shifted to 2.14 ppm. On the other hand, the di-substitution was more complicated and all properties were signed.

### 2.2. Bio-Evaluation

#### 2.2.1. Antibacterial Assay Using Agar-Well Diffusion Method

The antibacterial properties of chitosan and its alkyl derivatives have received a significant amount of interest in recent years. Furthermore, alterations in pathogenic microorganisms that make them resistant to antimicrobial materials have prompted scientists to develop novel antimicrobial materials with potential efficacy against new pathogenic microbes. The antimicrobial properties of chitosan and the two alkyl derivatives were first investigated using an agar-well diffusion technique against Gram-positive and Gram-negative bacteria (*S. aureus* and *B. cereus*) (*E. coli* and *P. aeruginosa*). Antimicrobial activity inhibition zones were measured, as shown in Table 2. Chitosan and its derivatives showed outstanding antibacterial activity against the microorganisms tested. Furthermore, the results showed that the two alkyl derivatives were significantly more effective against the indicator bacteria than the parent chitosan. We deduced from these results that di alkyl substitution (Di-NHD-Ch) had the maximum activity in the presence of all microorganisms tested, regardless of structure.

The antimicrobial action of chitosan may vary depending on several intrinsic factors, such as the chitosan source; the molecular weight, which influences penetration inside microorganisms; and the synthesis of new chitosan derivatives with novel properties, which usually improve chitosan’s antimicrobial action. Three main chitosan interactions with diverse bacteria have been proposed, differing depending on the cell wall structure and metabolic activity [19,36]. The first process is based on an electrostatic contact between the positive charge of chitosan’s amine groups (NH_3_^+^) and the negative charge of certain bacteria’s cell walls, which causes intracellular components to escape. The second process demonstrates how the molecular weight of chitosan affects its ability to penetrate microorganism nuclei and bind to DNA. As a result, mRNA expression will be suppressed, and protein synthesis will be prevented. The third mechanism is the chelating capacity of chitosan to metal ions such as Ca^2+^, Mg^2+^, and Zn^2+^, which are essential elements for microbial growth and metabolic pathways such as spore production in Gram-positive bacteria. These variables were achieved in the novel medium molecular weight chitosan derivatives. Thus, we propose that the primary methods might be combined to implement the antibacterial potency of the synthesized chitosan derivatives. Previous research said that methyl halide could be used to make a number of alkyl derivatives of chitosan that were more antimicrobial, and the new compounds had good antibacterial, antifungal, antiparasitic, and anticancer properties [37,38].

Furthermore, the interaction of the antibacterial polymer with cell wall membranes can be improved by increasing the hydrophobicity of the alkyl side chain of chitosan. This demonstrates the synergistic increase in chitosan activities when mono and di substitutions are used against the microorganisms studied compared to the prime chitosan. They calculated the MIC and bacteria- and fungal-killing abilities of the new chitosan derivatives based on what they learned from this study. Many alkyl derivatives were prepared and showed an improvement in chitosan activity against different bacterial strains. Yang et al. prepared a water-soluble N-alkylated disaccharide chitosan derivative and evaluated it against *Escherichia coli* and *Staphylococcus aureus*. He found that the activity depended on the kind of disaccharide and the degree of substitution [39]. The Rabae group, prepared and investigated the antimicrobial properties of a series of N-alkylated chitosan. They found that the most active derivative was N-(2,2-diphenylethyl) chitosan with EC50 values of 0.031 and 0.23 g L^−1^ against *B. cinerea* and *P. grisea*, respectively [40]. Additionally, methylated N-(4-N, N-dimethylaminocinnamyl) chitosan chloride (MDMCMCh), methylated N-(4-pyridylmethyl) chitosan chloride (MPyMeCh), and N,N,N-trimethyl chitosan chloride (TMChC) were prepared and tested against *Escherichia coli* ATCC 25922 (Gram-negative) and *Staphylococcus aureus* ATCC 6538 (Gram-positive) bacteria. The study demonstrated that the MDMCMCh showed higher antibacterial activity than TMChC, while MPyMeCh exhibited reduced antibacterial activity against both bacteria [41].

#### 2.2.2. Determination of MIC

The lowest concentrations of chitosan and its two alkyl derivatives (Mono-NHD-Ch and Di-NHD-Ch) that limit the development of the tested strains after overnight incubation were determined as minimum inhibitory concentrations (MICs) in the current study. This method is critical and focused on determining microorganism susceptibilities to our investigated materials and evaluating the antibacterial efficacy of these novel compounds. As shown in Table 3, the MIC of the two chitosan derivatives were assessed and compared with native chitosan. The results showed that the typical microbial growth inhibition characteristics were achieved, increasing the tested polymers’ antimicrobial activity. In addition, the results showed that the concentration (25 g/mL) of all polymers did not affect microbial cells. Still, the MIC of chitosan and the novel formulations was judged to be 50 g/mL, except in the case of *B. cereus*, which required 100 g/mL to halt their growth. Furthermore, pure chitosan did not affect *S. aureus* at 50 g/mL and required 100 g/mL to do so. Thus, in general, the antibacterial response of the di alkyl substitution is better than that of the mono substitution against all tested microorganisms.

Di-NHD-Ch at 50 g/mL showed maximum inhibition ratios of 48.18%, 39.23%, and 24.37% against *E. coli*, *P. aeruginosa*, and *S. aureus*, respectively, compared to Mono-NHD-inhibition Ch’s percentages of 41.37%, 27.1%, and 24.37% against the same tested microorganisms. At 100 g/mL, both tested derivatives began to inhibit *B. cereus*, with Mono-NHD-Ch exhibiting a 39.27% inhibition rate and Di-NHD-Ch showing a 43.54%inhibition rate. The inhibition ratios of microbial strains rose as the concentrations of chitosan and its derivatives increased, reaching a maximum at the highest measured concentration (250 g/mL). Accordingly, Di-NHD-Ch at a 250 g/mL dosage inhibited *E. coli*, *S. aureus*, and *B. cereus* almost wholly, with percentages ranging from 97 to 99%. In contrast, the most significant inhibition ratio against *P. aeruginosa* was 90%. However, using pure chitosan and Mono-NHD-Ch, these complete microbial growth inhibitions were not achieved. This behavior can be explained by the di substation derivative’s increased hydrophobicity, which makes it easier for it to stick to bacterial cell walls and start killing them.

In previous research, pure chitosan had 125 and 500 g/mL MIC values against *S. aureus* and *E. coli*, respectively. Against the same bacterial strains, the modified chitosan hydrogel and oxalyl bis 4-(2,5-dioxo-2H-pyrrol-1(5H)-yl) benzamide had MICs of 125 and 3.91 g/mL, respectively [42]. Another study found that the MIC of O-quaternary ammonium chitosan (OQCS) ranged from 250 to 600 g/mL. On the other hand, O-quaternary ammonium N-acyl thiourea chitosan (OQCATUCS) had more significant antibacterial activity than OQCS and chitosan, with MICs ranging from 125 to 250 g/mL [43]. It is worth noting that the MIC of our produced materials was in the range of 50–100 g/mL, with significant activity, and 250 g/mL of Di-NHD-Ch exhibited complete inhibition of Gram-positive bacteria and potent inhibition of Gram-negative bacteria up to 90%. Our findings suggest that modifying pure chitosan with different active groups might enhance and extend its antibacterial action, as demonstrated in prior studies. The difference in cell wall structures between Gram-positive and Gram-negative bacteria, which is explicitly shown at 250 g/mL, is due to the more significant activity of Di-NHD-Ch on Gram-positive bacteria. These characteristics were also found when Mono-NHD-Ch was applied to the investigated bacterial strains, but at a lower activity level. These antibacterial strategies are in line with earlier findings [44]. Two factors can explain this: First, Gram-negative bacteria have three barrier membranes that prevent materials from entering the cells, including a hydrophobic outer membrane, peptidoglycan, and cell membrane, whereas Gram-positive bacteria have a thick peptidoglycan that includes negatively charged teichoic acid molecules. Gram-positive bacteria’s structures enhance their propensity for interacting with chitosan, which has a positive charge, causing bacterial cell destruction. Second, the presence of porin channels within Gram-negative bacteria’s outer membrane may prevent chitosan residues from entering the cells. However, at a concentration of 50 g/mL, the inhibition ratios of Di-NHD-Ch toward Gram-negative bacteria are higher than for Gram-positive bacteria, implying that at high concentrations, the highest level of derivative adheres firmly to the cell wall and outer surface via electrostatic interaction, and an excess of polymer enters the cells to inhibit protein synthesis. Furthermore, the well diffusion approach supports our hypothesis of antibacterial activity. As a result, the antibacterial properties of the produced chitosan derivatives may be attributed to the two processes outlined in the preceding section, which are dependent on interactions with cell wall structure and the suppression of protein synthesis via cell DNA binding.

#### 2.2.3. Bactericidal Behavior

This research is critical in determining how bacteria react to novel medicinal drugs to minimize doses, intervals, and duration. Several parameters can influence this research, including material concentration, growing circumstances, bacterial density, and time. In this study, 150 g of native chitosan and its two alkyl derivatives, Mono-NHD-Ch and Di-NHD-Ch, were tested for time intervals against the pathogenic organisms previously stated. The findings of the bactericidal experiment are depicted in Figure 4, which demonstrates the relationships between the microbe inhibition rate (%) and contact duration in the range of 0–6 h. Based on the types of tested polymers and microorganisms, different performances of bacterial strains may be shown in the figures. However, both chitosan and its derivatives demonstrate bacteriostatic behavior in E. coli, with virtually identical inhibitory percentages of 54–60% during the studied experimental time. This might be because *E. coli* is very resistant to most common antibiotics, so it needs a combination of medicines to control growth and stop biofilms from forming, which can cause cystic fibrosis and even death in people.

On the other hand, the *P. aeruginosa* profile shows different behavior according to the tested polymer. Chitosan started to exhibit bacterial inhibition of 55.8% and showed a notable resistance after 4 h. The inhibition rate was 62–67% for the first 3 h, and after that it decreased to 35% at 6 h. Di-NHD-Ch shows the most stable inhibition, which refers to the high sensitivity of *P. aeruginosa* to Di-NHD-Ch compared to other tested polymers.

The interaction of the tested polymer shows a significant increase in sensitivity against *S. aureus* for the first 3 h, up to 71.2%. The bacterial strain offers resistance after that, showing inhibition activity of around 38% after 6 h. On the other hand, the Gram-positive bacteria *B. cereus* demonstrates a relatively stable sensitivity that slightly increases in Mono-NHD-Ch.

#### 2.2.4. Cytotoxicity

Table 4 and Figure 5 demonstrate that the viability of Wi-38 is sustained above 98% after incubation with Ch and Mono-NHD-Ch up to 1 mg and 0.5 mg, respectively. At 1 mg of Mono-NHD-Ch, cell proliferation was reduced slightly to 94%. In contrast to Ch and Mono-NHD-Ch, 1 mg, 0.5 mg, and 0.25 mg of Di-NHD-Ch caused severe cell activity decreases of less than 50%, 74%, and 82%, respectively. The viability was 92.64% and 99.09%, respectively, at the two lowest-used doses of Di-NHD-Ch. Figure 6 shows no marked alteration in Wi-38 cell morphology after incubation with 0.5 mg of Ch and Mono-NHD-Ch, while this same dose of Di-NHD-Ch caused moderate damage. These findings indicate that Di-NHD-Ch was more cytotoxic than Di-NHD-Ch and that Mono-NHD-Ch and Di-NHD-Ch were safe up to 1 mg and 0.125 mg, respectively.

## 3. Materials and Methods

### 3.1. Materials

1-Bromohexadecane was obtained from Sigma-Aldrich (Taufkirchen, Germany), 1-Methyl-2-pyrrolidinone (Purity 99%, M.W., 99) was obtained from Alfa Aesar (Karlsruhe, Germany), Chitosan (MW. 100,000–300,000 Dalton) was obtained from Across Organics (New Jersey, USA), hydrochloric acid (purity 37%) and sodium hydroxide pellets (purity 99–100%) were purchased from Sigma-Aldrich (Taufkirchen, Germany). Sulfuric acid (purity 98%, M.W., 96) was purchased from Sigma-Aldrich (Taufkirchen, Germany) and Ethanol (purity 99.9%, M.W.46.07) from International Co. for Supp. & Med. Industries (Greater Cairo, Egypt). Luria Bertani Broth (LB) was purchased from HIMEDIA LABORATORIES, Mumbai, India

#### Microorganisms

Wild bacteria were isolated and identified from hospitals in Alexandria, Egypt, according to our published protocol. All collected bacteria showed significant resistance to the multidrug antibiotics [45,46].

The developing materials’ antibacterial and antifungal performance were tested against Gram-negative (*Escherichia coli* (KY550380) (*E. coli.*) and *Pseudomonas aeruginosa* (KY550378) (*P. aeruginosa*)) and Gram-positive (*Staphylococcus aureus* (KY421197) (*S. aureus*) and *Bacillus cereus* (*B. cereus*)) bacteria. All strains were obtained by growing overnight in LB broth medium containing glycerol stocks at 37 °C and 150 rpm (peptone 1%, yeast extract 0.5%, NaCl 1%).

### 3.2. Methods

#### 3.2.1. Preparation of N-Alkyl Chitosan Derivative

The production of mono and di N-alkyl substituted chitosan followed the procedure with minimal changes [47], as illustrated in Figure 1. First, N-Alkyl chitosan was synthesized by the reductive alkylation of chitosan using 1-Bromohexadecane in the presence of a strong base (NaOH) at 60 °C. In a two-necked flask set in a constant temperature water bath at 60 °C, 2.0 g of chitosan and 4.8 g of sodium iodide (NaI, a catalyst) were dissolved in 80 mL of N-methyl-2-pyrrolidinone (NMP, a base) and swirled continuously until the chitosan was dissolved entirely. The condensation column was then attached to the flask. The flask was then filled with 11 mL of 15% (*w*/*v*) aqueous NaOH solution, followed by 1-Bromohexadecane, and the solution was stirred for another 6 h at 60 °C. Next, the mono N hexadecanyl chitosan (Mono-NHD-Ch) was precipitated using 200 mL of ethanol, centrifuged, washed twice with acetone on a sintered glass filter, and dried under reduced pressure. The steps were repeated starting from (Mono-NHD-Ch) to prepare the di substitution (Di-NHD-Ch). The products were weighed, and the yield of the reaction was calculated as 62.34% for Mono-NHD-Ch and 51.34% for Di-NHD-Ch according to the following formula:(1)$$reaction\;yield=\frac{actual\;product\;weight\;}{theortical\;weight}\times 100$$

#### 3.2.2. Characterization of Prepared Materials

Water uptake of chitosan and its alkyl derivative samples was determined as a weighed sample (0.06 g) of each tested material and was soaked in distilled water and allowed for equilibrium (up to 24 h). Each sample was then filtered off and carefully bolted with filter paper and with paper tissue that absorbed the excess of adsorbed water on the surface, and each sample was then weighed. Finally, the water swelling was determined by applying the following equation:(2)$$Water\;uptake\left(\%\right)=\left[\frac{M-Mo}{Mo}\right]\times 100$$
where *M* is the weight of the swollen sample and *Mo* is the weight of the dry sample, respectively.

Samples of chitosan and its alkyl derivatives were weighed before and after 3 h of drying at 105 °C. The following formula was used to determine the amount of moisture in the sample:(3)$$Moisture\;content\;\left(\%\right)=\frac{\left[Mo-M\right]}{Mo}\times 100$$
where *Mo* is the sample’s weight before drying, and *M* is the sample’s weight after drying.

Ion exchange capacity was measured as a known weight of each test of chitosan or its alkyl derivatives dispersed in (20 mL) of 0.1 M H_2_SO_4_ solution. The mixture was soaked for 24 h, then filtered, and the aliquot was titrated against a standard solution of sodium hydroxide 0.1 M. Similarly, 20 mL of 0.1 M H_2_SO_4_ was tested as a control. The following equation was used to calculate the ionic capacity of chitosan and its alkyl derivatives samples:(4)$$Ion\;exchange\;capacity\left(\frac{meq}{g}\right)=\frac{\left(V_{2}-V_{1}\right)a}{w}\;\;\;\;\;\;\;\;\;\;\;$$

The volumes of NaOH required for titration of H_2_SO_4_ in the absence and presence of chitosan and its alkyl derivatives are *V_2_* and *V_1_*, respectively; a is the normality of NaOH, and w is the sample weight.

The functional groups of chitosan and its alkyl derivatives were verified using an FT-IR spectrophotometer (Shimadzu FTIR-8400S, Japan). An amount of 1−2 mg of samples were mixed and ground with 200 mg KBr before being scanned in the range of 4000 and 400 cm^−1^.

The electronic spectrum of the chitosan and chitosan derivatives was estimated using a spectrophotometer (Model Ultrospec 2000). An amount of 0.05 g of chitosan or chitosan derivatives were dissolved in 10 mL of 2% acetic acid solution under heating. The electronic absorbance of the solution in a quartz cell was scanned from 200–600 nm.

NMR spectral analysis: ^1^H NMR spectra of chitosan and chitosan derivatives were obtained using JEOL 500 MH NMR spectrometer, Japan, at 500.2 MHZ. Polymer solution was prepared by dissolved it in DMSO/TFA and was analyzed by a 5 mm ^13^C-^1^H dual probe head at 25 °C. The spectra were accumulated into 32 K data points and processed using exponential multiplication with 2 Hz line broadening into 128 K spectra. For the resulting spectra, 25,000–35,000 scans were accumulated. All spectra were accumulated under identical conditions using power-gated Waltz decoupling with a 25-degree measurement pulse and 1 s pre-pulse delay.

Morphological changes of the sample’s surface were analyzed using a secondary electron detector of SEM (Joel Jsm 6360LA), Japan.

Differential scanning calorimetric analysis of chitosan and chitosan derivative samples (5 mg in sealed Al-pan) was performed in a temperature range of ambient −500 °C at a heating rate of 10 °C/min under nitrogen flow (10 mL/min) using a differential scanning calorimeter device (Shimadzu DSC–60A, Japan). Morphological changes of the sample’s surface were analyzed using a secondary electron detector of SEM (Joel Jsm 6360LA), Japan.

#### 3.2.3. In-Vitro Evaluation

##### Antibacterial Evaluation Using Agar-Well Diffusion Method

The bacterial properties of chitosan and its alkyl derivatives were examined according to agar-well diffusion against *E. coli*, *P. aeruginosa*, *S. aureus*, and *B. cereus*, as described by the reported method [48,49]. For example, 50 µL of fresh bacterial cultures were spread on Luria Britani (LB) agar medium. Precisely 30 µL of chitosan and its alkyl derivatives were loaded into agar wells, 6 mm in diameter. The plates were incubated at 37 °C for 24 h. The width of the clear zone was then measured with a ruler. The test was repeated in triplicate.

##### Minimum Inhibitory Concentration (MIC) Determination

The effect of various doses of chitosan and chitosan derivatives on the growth of *E. coli*, *P. aeruginosa*, *S. aureus*, and *B. cereus* was determined using the microtiter plate technique [50,51]. First, overnight strain cultures were prepared by inoculating the strains in LB broth and incubating them in a shaking incubator at 37 °C and 150 rpm. Next, bacterial cultures were diluted 100 times in the same broth medium to achieve optical densities (0.9) at 600 nm for all species. Then, 20 µL of diluted cultures were placed onto a sterile 96-well microplate with various final concentrations of chitosan and chitosan derivatives (25, 50, 100, 150, 200, and 250 g/mL). Prior to usage, the tested materials were sterilized using a syringe filter (0.22 m). The wells were filled to 200 µL with LB broth, stirred for 2 min at 100 rpm with a bench shaker, and incubated for 24 h at 37 °C. To evaluate the turbidity of bacterial cultures, the microtiter plates were shaken for 30 s and monitored at 600 nm using a microplate reader. The experiments were duplicated and recorded in triplicate. The following equation was used to compute the percentages of microbial growth inhibition:(5)$$Inhibition\;percent\left(\%\right)=\frac{\left[Nornal\;activity-inhibited\;activity\right]\;}{normal\;activity}\times 100$$

##### Bactericidal Activity

A microtiter plate was used to test the bactericidal properties of chitosan and its derivatives. To acquire optical densities (1.8) at 600 nm for all cultures, the overnight cultures of various microorganisms were diluted with an autoclaved LB medium. After that, 1 mL of each cell suspension was combined with 1 mL of chitosan and chitosan derivatives at 1% (*w*/*v*). After incubation at 37 °C for various durations (0, 1, 2, 3, 4, 5, and 6 h), the samples were extracted, and 10 µL from each sample was inoculated in 96-well microplates, followed by filling the wells with LB broth up to 200 µL. The microplates were stirred and incubated for 24 h at 37 °C. All experiments were repeated three times, and the turbidities of incubated cultures were measured at 600 nm with a microplate reader, as previously indicated. The average and standard deviations were computed and recorded [52].

##### Cytotoxicity of Chitosan and Its Alkyl Derivatives

A human standard fetal lung cell line (Wi-38) was used to investigate the toxicity of chitosan and its alkyl derivatives (Mono-NHD-Ch and Di-NHD-Ch) according to the method described by Mosmann (1983) [53]. Human Wi-38 cells were grown in DMEM medium (Lonza, USA) containing 10% fetal bovine serum (GIBCO, USA). This cell line was sub-cultured for two weeks prior to the trypsin EDTA test. Trypan blue dye and a hemocytometer were used to determine their vitality and to count them. Wi-38 cells were seeded at 1 × 10^4^ cells per well in 96 well culture plates and incubated at 37 °C in a 5% CO_2_ incubator. Cells were treated with serial amounts of tested polymer samples after 24 h for cell attachment. After 72 h of incubation in a 5% CO_2_ incubator, 20 µL of MTT solution (5 mg/mL) was added to each well and incubated for 4 h at 37 °C in a 5% CO_2_ incubator. The MTT solution (Sigma, USA) was removed, and the insoluble blue formazan crystals trapped in cells were solubilized in 150 µL of 100% DMSO for 10 min at 37 °C. The absorbance of each well was measured with a microplate reader (BMG LabTech, Germany) at 570 nm to estimate cell viability.

## 4. Conclusions

In this study, two different N-alkyl derivatives of chitosan were prepared and characterized as promising antibacterial polymers by condensing them with 1-Bromohexadecane and obtaining mono and di substitution (Mono-NHD-Ch and Di-NHD-Ch). The obtained results show impressive changes in their properties, which can be summarized as follows:

Physicochemical characterization of the new derivatives, such as ion exchange capacity and water uptake, was significantly changed;The N-alkyl derivatives were confirmed by the shift of the polymer characterization peak in their electronic spectrum and the alky signals at the HNMR chart;An FT-IR analysis of the prepared derivatives exhibited a significant new peak;The thermal stability of the chitosan derivatives was enhanced in the new derivatives;The antibacterial evaluation demonstrated that this new N-Alkyl derivative of chitosan exhibited better antibacterial activity than chitosan.

## Figures and Tables

**Figure 1 polymers-14-04011-f001:**
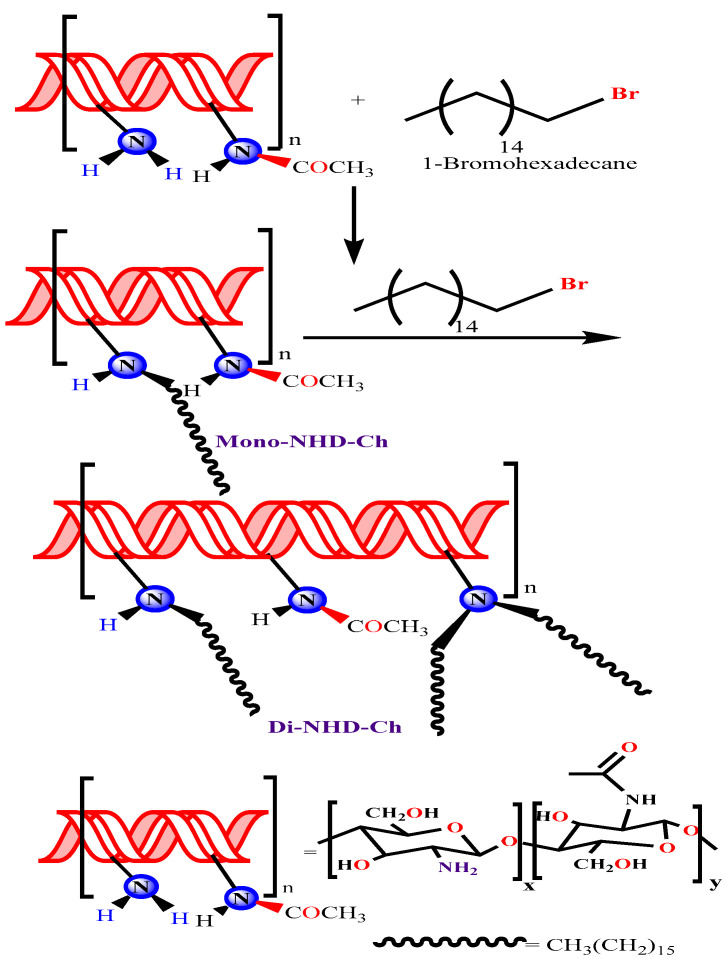
Schematic preparation of mono and di N hexadecanyl chitosan.

**Figure 2 polymers-14-04011-f002:**
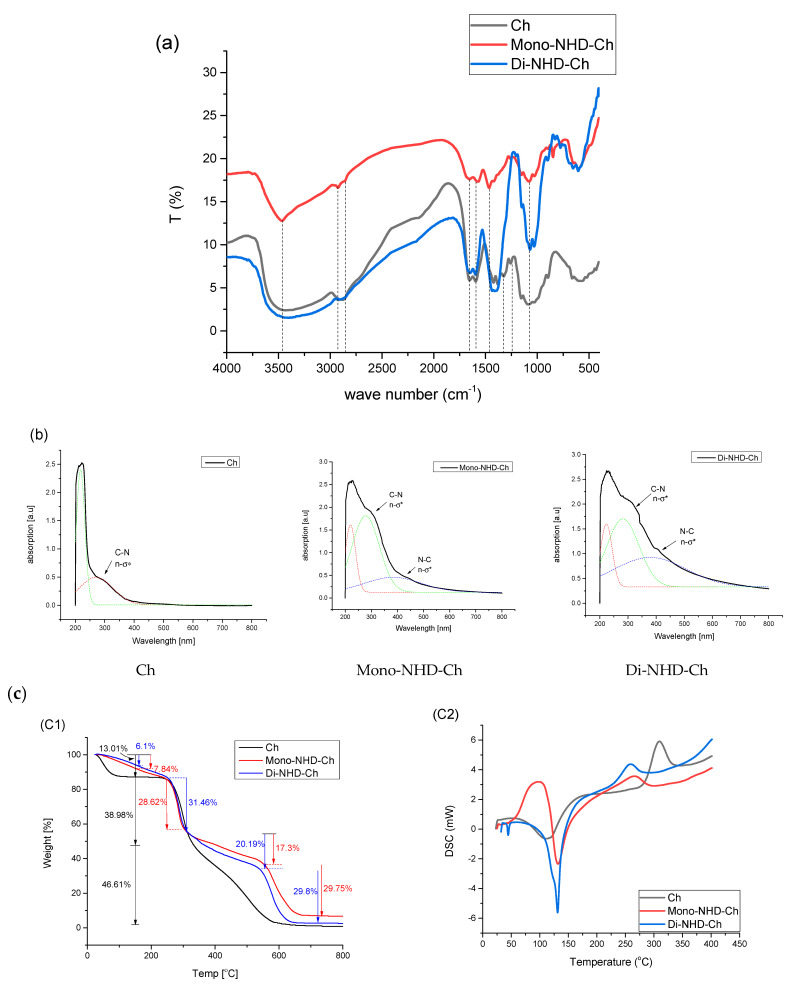
(**a**) FT-IR of chitosan and its alkyl derivatives. (**b**) Electronic spectra of chitosan and its alkyl derivatives. (**c**) Thermal analysis of chitosan and its alkyl derivatives: (**C1**) TGA analysis and (**C2**) DSC analysis. (**d**) SEM images of chitosan and its alkyl derivatives.

**Figure 3 polymers-14-04011-f003:**
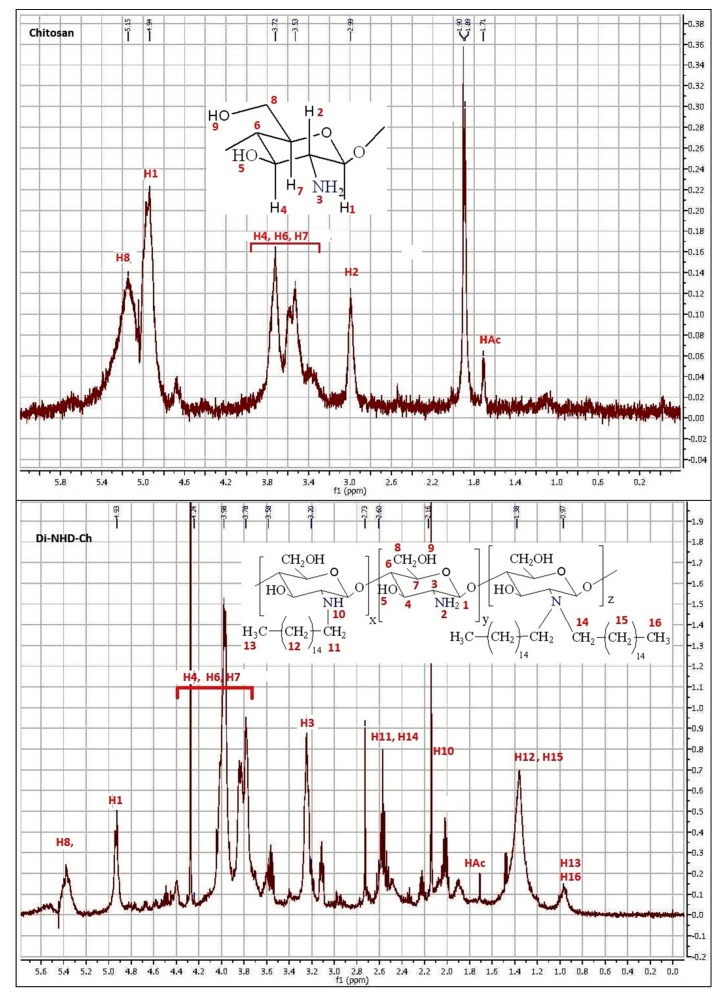
H NMR analysis of chitosan and its alkyl derivatives (Mono-NHD-Ch and Di-NHD-Ch).

**Figure 4 polymers-14-04011-f004:**
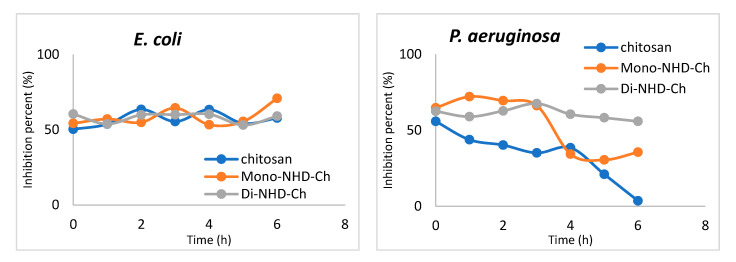
Bactericidal activity of chitosan and its alkyl derivatives (Mono-NHD-Ch and Di-NHD-Ch) against *E. coli*, *P. aeruginosa*, *B. cereus*, and *S. aureus*.

**Figure 5 polymers-14-04011-f005:**
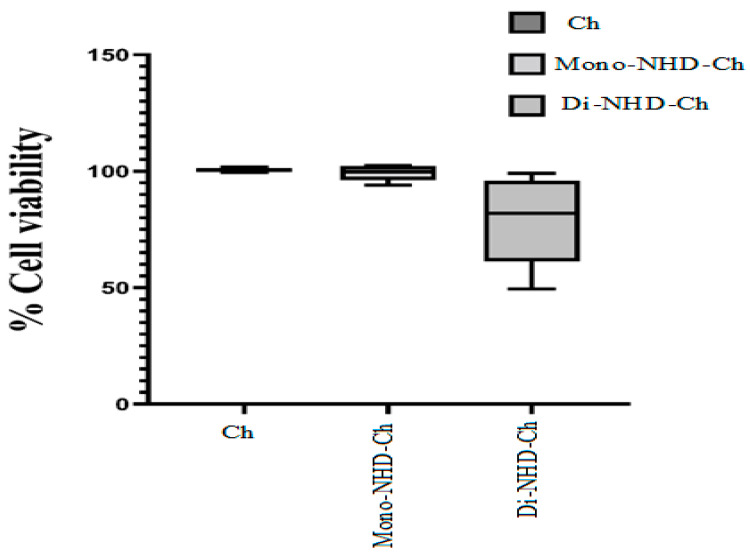
The percentage of Wi-38 cell viability after 72 h incubation with serial doses of Ch, Mono-NHD-Ch, and Di-NHD-Ch.

**Figure 6 polymers-14-04011-f006:**
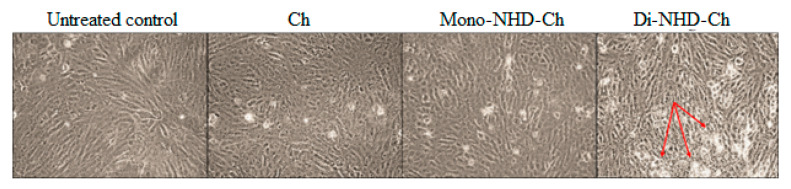
Wi-38 cell morphology after 72 h incubation with 0.5 mg of Ch, Mono-NHD-Ch, and Di-NHD-Ch. Arrows refer to cell damage.

**Table 1 polymers-14-04011-t001:** Elemental analysis of Chitosan and its Alkyl derivatives.

Sample Name	N%	C%	H%	C/N%	C/H%
Ch	7.17	39.05	7.21	5.45	5.42
Mono-NHD-Ch	2.37	20.41	6.64	8.63	3.07
Di-NHD-Ch	2.09	21.89	5.20	10.46	4.21

**Table 2 polymers-14-04011-t002:** Inhibition indices of chitosan and chitosan alkyl derivatives against *E. coli*, *P. aeruginosa*, *S. aureus*, and *B. cereus*. Values are expressed as mean ± SD (n = 3).

	Inhibition Zone (mm) in Diameter		
	Ch	Mono-NHD-Ch	Di-NHD-Ch
*E. coli*	15.8 ± 0.79	27.3 ± 1.4	29.1 ± 1.5
*P. aeruginosa*	13.5 ± 0.68	15.3 ± 0.77	21.1 ± 1.05
*S. aureus*	15.7 ± 0.78	19.8 ± 0.98	24.6 ± 1.23
*B. cereus*	16.4 ± 0.82	18.0 ± 0.9	23.3 ± 1.65

**Table 3 polymers-14-04011-t003:** MIC of chitosan and its alkyl derivatives (Mono-NHD-Ch and Di-NHD-Ch). (The data are the mean of three determinations. First, ± is standard deviation (SD)).

Sample Concentration (µg/mL)	Inhibition (%)											
	*E. coli*			*P. aeruginosa*			*S. aureus*			*B. cereus*		
	Ch	Mono-NHD-Ch	Di-NHD-Ch	Ch	Mono-NHD-Ch	Di-NHD-Ch	Ch	Mono-NHD-Ch	Di-NHD-Ch	Ch	Mono-NHD-Ch	Di-NHD-Ch
25	0	0	0	0	0	0	0	0	0	0	0	0
50	18.23 ± 0.91	41.37 ± 2.07	48.18 ± 2.41	24.12 ± 1.21	27.1 ± 1.36	39.23 ± 1.96	0	16.37 ± 0.82	24.37 ± 1.22	0	0	0
100	28.78 ± 1.44	68.13 ± 3.41	71.23 ± 3.56	33.7 ± 1.69	35.31 ± 1.77	41.22 ± 2.06	18.37 ± 0.92	33.47 ± 1.67	39.71 ± 1.99	37.21 ± 1.86	39.27 ± 1.96	43.54 ± 2.18
150	37.12 ± 1.86	71.47 ± 3.57	79.38 ± 3.97	35.69 ± 1.78	40.4 ± 2.02	55.71 ± 2.79	32.76 ± 1.64	41.38 ± 2.07	51.37 ± 2.57	52.31 ± 2.62	57.31 ± 2.87	74.27 ± 3.71
200	56.27± 2.81	79.21 ± 3.96	92.37 ± 4.62	40.01 ± 2.0	52.34 ± 2.62	78.16 ± 3.91	49.38 ± 2.47	49.31 ± 1.47	72.39 ± 3.62	58.23 ± 2.91	61.35 ± 3.07	91.32 ± 4.57
250	61.37 ± 3.07	82.52 ± 4.13	98.92 ± 4.95	49.67 ± 2.48	61.32 ± 3.07	90.06 ± 4.50	56.27 ± 2.81	63.87 ± 3.19	99.7 ± 4.99	61.35 ± 3.07	72.31 ± 3.62	97.45 ± 4.87

**Table 4 polymers-14-04011-t004:** The percentage of Wi-38 cell viability after 72 h of incubation with serial doses of Ch, Mono-NHD-Ch, and Di-NHD-Ch.

Weight (mg)	Ch		Mono-NHD-Ch		Di-NHD-Ch	
	Mean	SEM	Mean	SEM	Mean	SEM
**1**	99.473	3.389	94.051	1.130	49.473	3.087
**0.5**	99.783	0.217	98.270	0.380	73.159	4.795
**0.25**	100.615	0.615	99.832	0.169	81.926	2.295
**0.125**	100.936	0.924	101.930	0.600	92.642	2.982
**0.0625**	101.695	0.605	102.500	0.490	99.096	0.534

All data are expressed as mean ± SEM.

## Data Availability

Not applicable.

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
