# Peer review of "Preparation, Characterization, and Bio Evaluation of Fatty N- Hexadecanyl Chitosan Derivatives for Biomedical Applications"

_polymers, 2022, doi:10.3390/polym14194011_

Round 1

Reviewer 1 Report

(1) Please double-check if Table 3 is missing SD.

(2) I would recommend to improve the schematic presentation of Figure 1 for the reader to understand the chemistry and molecular models easier.

Author Response

Thank you for the valuable comments that we used to improve the manuscript. 

we answer all your comments in the attached file.

Best regards 

Reviewer 2 Report

The objective of this study was to improve the antibacterial activities of chitosan via N-alkyl substitution using 1-bromohexadecane. The investigations are interesting and the paper could be published after revision.   -The antibacterial properties were evaluated against Escherichia coli, Pseudomonas aeruginosa, Staphylococcus aureus, and Bacillus cereus. Why the microorganisms were investigated ? -The antibacterial properties of the novel alkyl derivatives increased substantially higher than chitosan. How this could be explained scientifically? - Antibacterial efficiency of the novel derivatives should by compared in conclusions with other derivatives used in this field of application. -Advantages and disadvantages of the developed derivatives should be compared in conclusions with that of other antibacterial materials, which are described in literature. -Thermo-gravimetric analyses (TGA) should be done for the materials before DSC measurements. It is not clear thermal degradation in DSC without the TGA. -Yields of the synthesized materials should be described. - Conclusions of the paper should be modified. At the moments just results are shortly described in the conclusions.

Author Response

Thank you for the valuable comments that we used to improve the manuscript. 

we answer all your comments in the attached file 

Best regards 

Round 2

Reviewer 2 Report

If editor and other reviewers agree that the paper is suitable for this journal I would recommend the paper for publication after the revision.